**Variations in return value estimate of ocean surface waves - a study based on measured buoy data and ERA-Interim reanalysis data**

T. Muhammed Naseef and V. Sanil Kumar*
Ocean Engineering Division, CSIR-National Institute of Oceanography (Council of Scientific & Industrial Research), Dona Paula 403 004, India
*Correspondence to*: V.Sanil Kumar (email:sanil@nio.org   Tel: 0091 832 2450 327 Fax: 0091 832 2450 602)

**Abstract.** An assessment of extreme wave characteristics during the design of marine facilities not only helps to ensure its safety but also the economic aspects. In this study, return levels of significant wave height (Hs) for different periods are estimated using Generalized Extreme Value (GEV) and Generalized Pareto Distribution (GPD) based on the waverider buoy data spanning for eight years and the ERA-Interim reanalysis data for 38 years. The analysis is carried out for wind-sea, swell and total Hs separately for buoy data. Seasonality of prevailing wave climate is also considered in the analysis to provide return levels for short-term activities in the location. The study shows that Initial Distribution Method (IDM) underestimates return levels compared to that of GPD. Maximum return levels estimated by GPD corresponding to 100 years is 5.10 m for monsoon season (JJAS), and corresponding pre-monsoon (FMAM) and post-monsoon (ONDJ) values are 2.66 m and 4.28 m respectively. Intercomparison of return levels by block maxima and r-largest method for GEV theory shows that maximum return level for 100 years is 7.20 m by r-largest series followed by monthly maxima (6.02 m) and annual maxima (5.66 m) series. The analysis is also carried out to understand the sensitivity of the number of observation for GEV annual maxima estimates. It indicates that the variations in the standard deviation of the series caused by changes in the number of observation are positively correlated with the return level estimates. The 100-year return levels of Hs by using GEV method give comparable results for short-term (2008 to 2016) buoy data (4.18 m) and the long-term (1979 to 2016) ERA-Interim shallow data (4.39 m). The 6-h interval data tend to miss high values of Hs and hence there is a significant difference in the 100-year return level Hs obtained using 6-h interval data compared to data at 1/2-h interval. The study shows that a single storm can cause large difference in the 100-year Hs value.

Keywords: surface waves, return period, Extreme value distribution, design wave height, north Indian Ocean

# 1. Introduction

Coastal zones are relatively dynamic than rest of the regions due to numerous natural as well as anthropogenic activities. Events such as extreme waves, storm surges, and coastal flooding make large catastrophes in the coastal region. The long-term (climate) behavior of sea state variables can be studied using non-stationary multivariate models that represent the time dependence of the variables (Solari and Losada, 2011). Various marine activities like the design of coastal and offshore facilities, planning of harbor operations, and ship design require detailed assessment of wave characteristics with certain return periods (Caires et al., 2005; Menéndez et al., 2009; Goda et al., 2010). Generally, Extreme Value Theory (EVT) is used for determination of return levels by adopting statistical analysis of historic time series of wave heights obtained from various sources such as in-situ buoy measurements (eg.: Soares and Scotto, 2004; Mendez et al., 2008; Viselli et al., 2015), satellite data (eg.: Alves et al., 2003; Izaguirre et al., 2010), and hindcasted or reanalysis data by numerical models (eg.: Goda et al., 1993; Caires and Sterl, 2005; Teena et al., 2012; Jonathan et al., 2014). EVT consists of two type of distributions viz. Generalized Extreme Value (GEV) family which includes Gumbel, Frechet, and Weibull distributions (Gumbel, 1958; Katz et al., 2002) and Generalized Pareto Distribution (GPD) which incorporates Peak Over Threshold (POT) approach (Pickands, 1975; Coles et al., 2001).

GEV distribution by annual maxima (AM) observations (Goda, 1992) is one of the widely used methods in the Extreme Value Analysis (EVA). The main difficulty with using this method is that the unavailability of a reliable observation at a location of interest. To overcome the data scarcity, different alternatives has been used by various authors such as Initial Distribution Method (IDM) in which the distribution of data as such is used (Alves and Young, 2003), r-largest approach (Smith, 1986), where a number of largest observation from a block of period are considered rather than one observation used in AM method. POT method (Abild et al., 1992) gives a good number of observations available for the analysis. Although there have been various proposals to automate threshold selection, threshold estimation for the application of the POT method to a single sample is still not resolved (Solari and Losada, 2012; Solari et al., 2017). GPD is another class of distribution introduced by Pickands (1975) and is used by several authors like Caires and Sterl (2005) and Thevasiyani et al. (2014). Teena et al. (2012) and Samayam et al. (2017) have carried out the EVA of ocean surface waves in the northern Indian Ocean based on wave hindcast data and ERA-Interim reanalysis data.

Most reliable source of ocean wave data is buoy measurements, and it can be used for EVA (Panchang et al., 1999). In this paper, data from a directional waverider buoy located in the central western shelf of India is used. Seasonality is one of the important aspects of climate data and therefore, it should be incorporated in the EVA of waves especially in a region like the Arabian Sea. Seasonal analysis of the extremes helps for the planning of short-term marine activities like offshore explorations, maintenance of coastal facilities, etc. In the present paper, the EVA is carried out by following both the GEV and GPD methods considering wind-sea, swell and total significant wave height (Hs) separately. The IDM and POT methods are used for total wave height analysis, and block maxima (annual and monthly maxima) and r-largest method are used in wind-sea and swell height analysis. Since the measured buoy data is for a short period of 8 years, the ERA-Interim reanalysis data from 1979 to 2016 is also used for comparing Hs value with 100-year return period.

The paper is organized as follows. Section 2 deals with data and methodology used in the analysis. It also presents the threshold selection adopted in the study and Sect. 3 explains the results obtained in the analysis categorized into seasons using total Hs data and comparison of return level estimation by different GEV approaches using wind-sea and swell height data. A case study is also included in the section for realizing the uncertainty related to observations in AM approach when limited number of observations are available. The influence of length of wave data on estimated Hs return value is also covered under this section. Section 4 provides the concluding remarks.

**2. Data and Methodology**

2.1 Data

Data used in the analysis is from Datawell directional waverider buoy deployed off Honnavar (14.304$^o$N; 74.391$^o$E) at a water depth of 9 m. The half hourly sampled data covers the period from March 2008 to February 2016. The waves at the location show strong intra-annual variations due to the prevailing wind system during monsoon and non-monsoon seasons (Sanil Kumar et al., 2014). To understand the local and remote influences on the design wave characteristics, we analyzed Hs of wind-sea, swell and total waves separately. The season wise study is also carried out since it will provide insight to the design wave heights for short-term coastal activities.

The Hs data from the ERA-Interim (Dee et al. 2011), the global atmospheric reanalysis product of the European Centre for Medium Range Weather Forecast (ECMWF) from 1979

to 2016 (38 years) is also used to evaluate the 100 and 50 year return period wave height in the shallow (water depth ~20m) and deep water. The shallow region is close to the buoy location and the deep water location is at a water depth of ~4000 m (Table 1). ERA-Interim used in the study has a spatial resolution of 0.125° X 0.125° and a temporal resolution of 6 h.

2.2 Methodology

EVA is carried out by following GEV Distribution model and POT method in which exceedance over a reliable threshold wave height can be fit into GPD. In POT method, distribution of excess, *x,* over a threshold *u* is defined as:

$$F_u(y) = \Pr\{x - u \le x | x > u\} = \frac{F(x) - F(u)}{1 - F(u)} \tag{1}$$

Where y=*x-u*. Pickands (1975) shows that distribution function of excess, $F_u(y)$, for a sufficiently high threshold *u* converges to GPD having CDF as:

$$G(x; k, \alpha, \beta) = \{1 - \left(1 - k\frac{x-\beta}{\alpha}\right)^{1/k}\} \qquad k \ne 0 \tag{2}$$

$$= 1 - e^{-(x-\beta)/\alpha} \qquad k = 0$$

GEV has cumulative distribution function (CDF) as:

$$F(X) = \exp\left\{-(1 - k\left(\frac{X-\beta}{\alpha}\right)^{1/k}\right\}k \ne 0 \tag{3}$$

$$= \exp\left\{-\exp\left(-\frac{X-\beta}{\alpha}\right)\right\} k = 0$$

Where α is scale parameter in the range of α >0, β is the location parameter with possible values of -∞ < β <∞ and k is the shape parameter in the range of -∞ <k<∞. GPD can be further categorized into three distributions based on its tail features.When k=0, GPD corresponds to an exponential distribution (medium-tailed or Pareto I type) with mean α; when k>0, GPD is short-tailed also known as Pareto II type; when k<0, distribution takes the form of ordinary Pareto distribution having long-tailed distribution (also known as Pareto III type). Parameter estimation and statistical distribution fitting are carried out by using WAFO (Brodtkorb et al., 2000) developed by Lund University, Sweden.

The analysis is carried out by using the wind-sea, swell and total Hs data covering ~ 8 years (2008-2016). From the measured data, to separate the wind-seas and swells, the method proposed by Portilla et al. (2009) is used. The separation algorithm is on the assumption that,

the energy at the peak frequency of a swell cannot be higher than the value of a Pierson-Moskowitz (PM) spectrum with the same frequency. If the ratio between the peak energy of a wave system and the energy of a PM spectrum at the same frequency is above a threshold value of 1, the system is considered to represent wind-sea, else it is taken to be swell. A separation frequency $f_c$ is estimated following Portilla et al. (2009) and the swell and wind-sea parameters are obtained for frequencies ranging from 0.025 Hz to $f_c$ and from $f_c$ to 0.58 Hz, respectively. GPD method is used for seasonal analysis of different period data series. GEV method is used for inter-comparison of return level estimation among wind-sea, swell and resultant data sets by extracting different block maxima series viz. seasonal maxima which contain highest observations from each season; monthly maxima contain one highest observation from each month, and annual maxima. The parameters are estimated using PWM method since the data set duration is very limited, and PWM method holds good results compared to other methods such as Maximum Likelihood (ML) method (Hosking et al., 1985).

To study the uncertainties related to the length of the observation, we extracted 3, 6, 12 and 24 h data series from the half hourly original data and carried out EVA. Since the wave climate in the study location strongly characterized by the prevailing seasonal behavior of wind system, we took further consideration of uncertainties related to a seasonal aspect of wave climate by extracting three seasonal data, viz., pre-monsoon (FMAM), monsoon (JJAS) and post-monsoon (ONDJ) seasons.

The major drawback of EVA using block maxima method, especially the annual maxima (AM), is that it does not consider the significant amount of observations which are closely related to storm features of the data set. Those omissions of observation would make variations in the final results of EVA to a great extent especially in the case when EVA is done for a very limited data set. EVT is based on one of the hypothesis that the observations under consideration are independent and identically distributed (Coles et al., 2001). In the case of ocean wave observations, we can expect its identical status for a large extent. Since POT approach re-samples the data over a threshold value, establishing identical and independence among the re-sampled observation is a tedious task. A suitable combination of threshold and minimum separation time between the re-sampled observations must be taken into account to establish independence among the observations.

The average duration of tropical storms in the Arabian Sea is 2-3 days (Shaji et al., 2014). So, in the present analysis, we fixed minimum 48 hours of separation time in between two consecutive storm peaks to ensure the independence of data points for the analysis. Then selected a tentative threshold value in such a way that there must be at least 15 peak values per year on average. That resulted at least 120 data points in each sub data sets used for the seasonal analysis. The resulting data series are used in further POT analysis. Further adjustment of the threshold is carried by Sample Mean Excess (SME) plots and Parameter Stability plots (PS plot). From these plots, we selected probable four thresholds and fitted corresponding GPD. A final threshold value is chosen by analyzing results obtained in different Goodness of Fit (GOF) tests such as Kolmogorov-Smirnov (KS) test, Anderson-Darling (AD) test and Cramer-von Mises (CM) test (Stephens, 1974; Choulakian and Stephens, 2001).

The distributions used in the analysis is validated using graphical tools like Quantile-Quantile (Q-Q) plots and CDF plots. In addition to above graphical tools, we checked the reliability of chosen thresholds for POT method by using different GOF tests such as KS, AD and CM tests (Table 2). p-value>0.05 indicates the selected distribution does not show a significant difference from the original data within 5 % significance interval.

## 3. Results and Discussion

### 3.1 Long-term statistical analysis of total Hs

The mean wave climate at the study location is characterized by annual mean Hs of 1.04 m. Maximum Hs of the data during 2008-2016 is 4.70 m and the next highest Hs is 4.34 m (Figure 1), whereas the highest wind-sea, and swell Hs are 4.29 m and 4.28 m respectively. Statistical analysis of Hs was carried out by considering the seasonal characteristics of the wave climate. To study the seasonal aspects of the return level estimation, the data is grouped into three different seasonal series, viz. FMAM, JJAS and ONDJ seasons in addition to full-year data. Since the study location is located off the central west coast of India, the wave climate shows distinct variability throughout a year. Previous studies like Anoop et al. (2015) reported that average Hs attains its peak around 3 m during JJAS and FMAM season is relatively calm (0.5-1.5 m) compared to that (1.5-2 m) in ONDJ. The seasonal analysis is carried out using Hs data following both the GEV and GPD methods. Here, Initial Distribution Method (IDM) is considered in GEV method rather than block maxima (Mathiesen et al., 1994). One of the challenging tasks for GPD modeling is the selection of a

suitable threshold value. The threshold should be high enough for observations to be independent and data after POT must have enough number of observations left in order to converge POT into GPD. SME plots and PS plots are used to select a range of initial thresholds. On analyzing the resultant GPD fit for those thresholds, final thresholds are chosen by the help of GOF tests which are presented in Table 2. Figure 2 and Table 3 shows the estimated parameters using PWM method for both GEV and GPD. It is clear that shape parameters in both cases are negative indicating the models are Type III distribution for GPD and Weibull distribution for GEV respectively. Table 3 also shows the RMSE in the chosen model for each data series with estimated CDF. It is evident that JJAS season has lesser RMSE (~0.07 m on average) when considering GPD model. While in the case of GEV model, full year data series has lesser RMSE (~0.02 m on average). ONDJ season shows a higher discrepancy in both cases resulting in average RMSE of 0.31 m and 0.54 m for GPD and GEV respectively. Figure 3 shows a typical SME and PS plots used for choosing a range of thresholds before fixing final threshold for POT analysis on each series. In this particular case (6 h data series of FMAM season) a range of thresholds from 1.10 m to 1.32 m were selected, and the final threshold of 1.19 m was fixed on analyzing the GOF test results (Table 2).

3.1.1 Full year

Here, we considered full year data series without dealing with seasonality and both the GEV and GPD are used in the analysis. Initially, a range of thresholds from 2.5 to 3.4 m was selected, and further adjustment of the threshold is carried out by analyzing the GOF test results. Table 2 shows the selected thresholds and corresponding GOF test results for each series in the full year data analysis. It is clear that the selected thresholds are in good agreement with GOF test results. Both KS test and CM test gives p-value > 0.32. Moreover, both CDF plots and Q-Q plots (see Figure 4: first and second rows, respectively) show selected GPD models made a good performance for the particular POT series. After acquiring best fit model, return levels (Table 4) were estimated for 10, 50 and 100 years. The GPD model estimates 10-year return level smaller than that of the maximum measured total Hs value by an extent of 5 to 15 %. Underestimation of 10 to 25 % from the maximum measured value was reported by Samayam et al. (2017) compared to the 36-years and 30-years return levels based on ERA-Interim reanalysis data for deep waters around Indian mainland. The initial distribution approach underestimates the return levels such a way that even 100 years return level does not cross the highest observation (4.70 m) in the data and the largest 100

years return level is reported as 4.26 m when dealing with half-hourly data series (Table 4). The large number of observations having very low Hs in the data series used in the analysis leads to the underestimation in the initial distribution method. Whereas, GPD model estimated 4.73 m and 4.96 m as 50 and 100 years return levels respectively. When considering different time interval data, both 12 and 24 h data series estimates lower return levels compared to other series by GEV model. It is evident that there are uncertainties related to the sampling interval adopted for the return value estimation. The standard deviation for GPD estimation when considering different time intervals is 0.57 m which are highest among the other seasonal data. GEV estimation reports even lesser spreading of return levels with 0.16 m standard deviation.

### 3.1.2 Pre-monsoon season

The data of February to May constitute the pre-monsoon data set. Pre-monsoon is the calmest season in the study location with maximum and average Hs of around 1.94 m and 0.73 m. Using SME and PS plots, a range of thresholds from 1.19 m to 1.32 m is selected for each time series and fitted corresponding GPD by using resultant POT. The final threshold selected by the help of GOF tests is presented in Table 2. KS and CM tests give p-value more than 0.43 and 0.45 respectively on an average (Table 2). Since the p-values are more than 0.05, the chosen POT is not significantly different from the time series data. CDF plots and Q-Q plots (Fig. 5) for the different data series of the season illustrate the reliability of the chosen model. Return levels for different return periods using a particular GPD are presented in Table 4. GEV estimation exhibit same characteristics of underestimation as shown in the full year analysis. Average 100 years return levels estimation using different time interval data using GEV model attained only 1.77 m which are less than the highest observed data point in the season, whereas, GPD reports average 100 years return level of 2.49 m. Time interval analysis for the season exhibits least discrepancies among the return level estimations compared to other seasons. Standard deviations of 0.11 m and 0.08 m for GPD and GEV estimations respectively were observed for 100-year return levels considering different time series data.

### 3.1.3 Monsoon season

Monsoon season data set covers observations from June to September and this season is characterized by rough wave climate at the study location. Hs of 4.70 m and 1.77 m are recorded as maximum and average during the season. A range of thresholds (2.78 to 3.49 m) are selected for preliminary GPD fitting as a result of interpreting SME and PS plots of each

data series, and corresponding final thresholds were selected after clarifying with GOF test results (Table 2). Both KS and CM tests report p-value > 0.56 indicating that the resulting POT for selected threshold converges into GPD. CDF and Q-Q plots in Fig. 6 shows the credibility of the adopted threshold value. Return levels for distinct return period were estimated using resultant POT. Table 4 provides 10, 50 and 100 years return period values estimated using GPD and GEV models. For half hourly data, GPD projects 4.80 m as 100-year return level, whereas GEV underestimates to 4.29 m. While considering different time interval data, GPD model shows 0.36 m standard deviation among the return levels for different time interval data. Both the 12 and 24 h series gave lower return levels compared to other series.

### 3.1.4   Post-monsoon season

Post-monsoon season constitutes data from October to January months of the year and the observed maximum Hs in this season is 2.41 m.  The majority of observations during this season lies below the average value of Hs. Only 32 % of the observations lie above 1.13 m and 8 % of the data are above 1.5 m. Hence, selecting the best threshold for the season was more difficult. GPD was fitted for a range of thresholds (0.7 to 1.3 m) selected from SME and PS plots corresponding to each series. Most suitable thresholds were selected after checking the goodness of GPD (Table 2). The GOF test results show that the ONDJ series holds maximum uncertainties on threshold selection due to lower p-values for KS test ranges from 0.13 to 0.48 and 0.19 to 0.45 for CM test respectively. Figure 7 shows the CDF and Q-Q plots. GEV and GPD estimation for post-monsoon season show very large difference among return levels (Table 4). The average percentage difference between 100 years return values obtained from GEV and GPD estimations is ~60%. It shows that GEV model clearly underperforms during ONDJ season when initial distribution methods were adopted. Highest return level reported by GPD model is 4.28 m, whereas GEV estimated about 2.3 m for the season. ONDJ accounts standard deviation of 0.30 m and 0.13 m for GPD and GEV estimation, respectively, while using different sampling intervals.

### 3.2   Long-term statistical analysis of wind-seas and swells

In this section, we relayed on GEV method based on block maxima. For that purpose, we extracted total, wind-sea and swell Hs data into different block maxima viz. monthly, seasonal and annual maxima series. Two seasonal maxima series is considered in such a way that one includes highest two observations in a season and another one consist of highest observation from each season. So monthly maxima series includes 96 data points. Both

seasonal maxima series (seasonal maxima 1 and 2) consist of 24 and 48 data points respectively. Annual maxima series covers 8 data points. Table 5 shows the estimated return levels corresponding to various return periods. It is clear that both seasonal maxima series provides highest return levels for total Hs (6.56 m and 7.20 m) and swell Hs (5.95 m and 6.35 m), whereas wind-sea Hs is 6.16 m when annual maxima series is considered. The GEV-AM model shows underestimation of 10-year return level compared to the maximum measured data. The annual maxima series resulted in 5.66 m as the 100-year return level for the total Hs (Figure 8), which is comparable with Teena et al. (2012) estimation for the location off the central west coast of India.

We did a separate analysis of the annual maxima series to get insight into the abnormal results observed for wind-sea data series. Here, we considered four unique series of different length by taking annual maxima observations from 2008 to 2016. That is, first series (S1) consist of 5 data points (2008-2012) and second series (S2) consists of 6 data points (2008-2013) and so on. The density plots showing the probability for different wave height class is presented in Fig. 9 along with the corresponding GPD fit. We calculated the standard deviation for each series and the percentage difference between each series with the parent series (S0). The result shows that return levels are positively correlated with standard deviation (Table 6). In the case of total Hs, the correlation between the changes in standard deviation and the corresponding changes in 100-year return levels are 0.997, whereas for wind-sea and swell; it is 0.964 and 0.647 respectively. Annual maxima of wind-sea (4.29 m) for the year 2015 made an abrupt change in the standard deviation of the series by about 0.46 m which are more than 17 % of the average of the series excluding 2015. So, the 100 years return level for wind-sea overshoot for about 6.16 m making 66 % difference from return value obtained for S3 series. In this case study, the length of the special series under consideration does not influence on the estimated return levels. That is, in the case of total Hs series, 100 years return levels for S1 series is greater than both S2 and S3 series. Same characteristics can be seen in the case of swell Hs also. Therefore, return levels for annual maxima by GEV model have greater influence over how a single data point, i.e., the annual maxima, alter the standard deviation of the series rather than the changes in the length of the series.

3.3    Influence of length of wave data on estimated significant wave height return value

An analysis is carried out to check uncertainties in return level estimation related to the length of the wave record. From the 1/2 h buoy measured data, data at 6-h interval are extracted and used for the analysis and the return levels obtained by using 6-h measured buoy data are compared with the return level obtained from the 6-h ERA-Interim data at shallow and deep locations (Figure 8). Six hourly ERA-Interim reanalysis data for 38 years (1979-2016) is used in this analysis. Buoy data consists of 11479 data points and ERA-Interim data consists of 55520 data points (Table 1). The highest observed Hs in the 6 hourly buoy data is 4.11 m followed by 4.03 m. while maximum Hs in the ERA-Interim shallow is 5.45 m and in the ERA-Interim deep is 7.13 m. The Hs values at deep location is ~1.4 times the values at the shallow location and it resulted in higher return level of Hs at the deep location. Sanil Kumar and Naseef (2015) observed that ERA-Interim overestimates the Hs for shallow water locations along the west coast of India due to swell height overestimation and the difference between the ERA-Interim Hs and the buoy Hs is up to 15%. For the study location, the storm induced wave heights during the non-monsoon period are less than the monsoon induced waves. June first week is the onset of Indian summer monsoon and the maximum Hs in the study area is due to monsoon influence and in all years it is during June to September. The 100-year return levels by using GEV method give comparable results for buoy data (4.18 m) and ERA-Interim shallow data (4.39 m), while that for ERA deep is 5.67 m (Figure 8). It is clear that the 100-year Hs return level using GEV for ERA-Interim data is lower than the maximum Hs in the data, while in the case of buoy data, 100-year return level is slightly higher than the highest Hs value. The return levels obtained by GPD method shows significant discrepancy among 100-year estimates. The 100-year return level obtained for buoy data is 4.46 m, but that using ERA-Interim shallow data is 6.18 m and that for ERA-Interim deep is 7.28 m. The Hs return level for 100-year for deep water has closer values following GEV and GPD, while in the shallow water, a significant difference is obtained. The 6-h interval data tend to miss 18 values of Hs between 4.11 and 4.70 m and hence there is a significant difference in the 100-year return level of Hs based on GEV-AM obtained using this data compared to that based on the data at 1/2-h interval.

We have examined the difference in the return level of Hs by considering data in different blocks; i.e. 10, 20, 30 and 38 years using the ERA-Interim shallow water data. The study indicates a large underestimation (~18%) in the return level estimate if we consider only the first 10 years (1979-1988) data in place of the 38 years (Figure 10). The large difference in the values of Hs return level is due to the occurrence of a tropical storm in the Arabian Sea

during 9-12 June 1996, which resulted in a high wave heights, Hs up to 5.46 m, whereas the maximum Hs excluding this storm is 4.63 m. During the 1996 storm, Hs of 5.69 m is measured by a Datawell directional waverider buoy moored at 23-m water depth off Goa (Sanil Kumar et al., 2006), which is ~150 km north of the present study area. The data blocks containing this storm data i.e. the 20 years (1979-1998) and 30 years (1979-2008) data did not show much difference in the 100-year Hs value compared to the 38 years data. If we consider only the last 10 years (2007-2016), it resulted in 7% underestimation in the 100-year Hs value. The study shows that a single storm can create a large difference in the 100-year Hs value, compared to the differences in values resulted from different length of the data block.

The long-term and decadal trend of wave climate in the different parts of major oceans is studied (Young et al. 2011). We have examined the trend in Hs at the shallow location based on the ERA-Interim data from 1979 to 2016. The study shows that the annual maximum Hs shows a weak increasing trend (1.1 cm $y^{-1}$), whereas there is no significant trend in the annual mean value (Figure 11). Sanil Kumar and Anoop (2015) observed that during 1979 to 2012, the average trend of annual mean Hs for all the locations in the western shelf seas is 0.06 cm $y^{-1}$.

3.4    Influence of water depth on the measured buoy data

The relative water depth based on spectral peak period ($d/L_p$) indicates that most of the time (97.8 to 99.3%), the wave regime is in intermediate water (Table 7). Only during 0.1 to 0.8% of the time, the waves satisfy the deep water condition. Hence, the waves measured by the buoys are influenced by the bathymetry and the wave characteristics will be different in the deep water.  The wave rose plots during March 2008 to February 2016 based on the measured buoy data and the ERA-Interim reanalysis data at shallow and deep water locations are presented in Figure 12. As the waves move from deep to shallow waters, the direction of high waves shifted from southwest to west. The limiting value of wave height based on breaker criteria is 0.6 to 0.78 times the water depth (Massel, 1966). The maximum Hs in the measured buoy data is 4.70 m and some of the waves containing this record will arrive very steep or broken at 9-m water depth since the maximum wave height is 1.65 to 1.8 times the Hs.

**4. Conclusion**

Long-term statistical analysis of extreme waves is carried out based on GEV and GPD models using measured buoy data from March 2008 to February 2016 and the ERA-Interim data from 1979 to 2016. Return levels are calculated for resultant, wind-sea and swell Hs separately. The analysis is also conducted for data under three different seasons. The parent data are resampled into 3, 6, 12 and 24 hourly series and estimated the discrepancy in return level estimation. Selection of appropriate thresholds for POT method is justified using different GOF tests results. Analysis of the total Hs shows that IDM approach underestimates return levels for different seasons compared to corresponding GPD. The 100 years return level estimated by IDM are almost comparable with corresponding GPD estimation for ten years period, but there is a significant difference in the return level estimates when considering different sampling intervals. IDM estimates largely underestimates return levels for the post-monsoon season since the majority of the observation in this season lies away from its tail of the distribution.

Long-term statistics of wind-sea and swell data are calculated by GEV model following block maxima and r-largest methods. Annual maxima and monthly maxima are considered for block maxima series, and two seasonal maxima series are considered for the r-largest method. It is shown that these methods give higher return levels than GPD models. The r-largest method provides 7.20 m as 100-year return level when compared to 5.27 m of GPD model. The sensitivity analysis of GEV-AM model shows that change in the standard deviation of data series under consideration makes discrepancies in the return level estimates rather than a change in the length of the series. Both GEV and GPD models underestimate 10-year return levels compared to maximum measured data. The 100-year return levels by using GEV method give comparable results for short-term (2008 to 2016) buoy data (4.18 m) and the long-term (1979 to 2016) ERA-Interim shallow data (4.39 m). The 6-h interval data tend to miss high values of Hs m and hence there is a significant difference in the 100-year return level Hs obtained using this data compared to data at 1/2-h interval. The ERA-Interim data shows that from 1979 to 2016, the annual maximum Hs shows a weak increasing trend (1.1 cm y$^{-1}$). The study shows that a single storm can create a large difference in the 100-year Hs value, compared to the differences in values obtained from different length of the data block.

**Acknowledgments**

The Director, CSIR-National Institute of Oceanography, Goa, provided facilities to carry out the study. Shri Jai Singh, Technical Officer, CSIR-NIO, assisted in the data analysis. This work forms part of the Ph.D. thesis of the first author and is CSIR-NIO contribution number 2017 under the institutional project MLP1701. We thank the Editor Dr. Mauricio Gonzalez and the two anonymous referees for the suggestions for improving the manuscript.

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

**Figure captions**

Figure 1: Time series plot of the significant wave height measured by buoy and from ERA-Interim data at shallow and deep water.

Figure 2: Estimated shape parameters for different seasonal data with different sampling intervals used in a) GEV and b) GPD model.

Figure 3: A Typical (a) SME and (b) PS plots used for selecting a range of thresholds required for POT analysis. In this particular case, a range of 1.19 m to 1.32 m was selected.

Figure 4: Figure corresponding to full year analysis. (a) to (e) is CDF plots for ½ hourly to 24 hourly data respectively, sub-figures, (f) to (g) are corresponding Q-Q plots and (k) to (o) are corresponding return levels estimated using GPD model.

Figure 5: Same as in Figure 4 but corresponding to pre-monsoon season.

Figure 6: Same as in Figure 4 but corresponding to monsoon season.

Figure 7: Same as in Figure 4 but corresponding to the post-monsoon season.

Figure 8: Return levels of significant wave heights for different return periods based on buoy data (2008-2016), ERA-Interim shallow and deep water (1979-2016) at 6-h interval by GEV model using annual maxima series.

Figure 9: Density plots showing the probability for different wave height class. Total, wind-sea and swell Hs are presented in rows wise. Columns correspond to selected number of data points (5 to 8 years). The solid curve is the corresponding GPD fit.

Figure 10. Return levels of significant wave heights for different return periods based on ERA-Interim shallow water data in different block years by GEV model using annual maxima series

Figure 11. Variation of (a) annual maximum and (b) annual mean Hs at the shallow locations based on ERA-Interim data. The solid line indicates the trend in Hs during 1979 to 2016

Figure 12: Wave rose plots during March 2008 to February 2016 based on the measured buoy data and the ERA-Interim reanalysis data at shallow and deep water locations

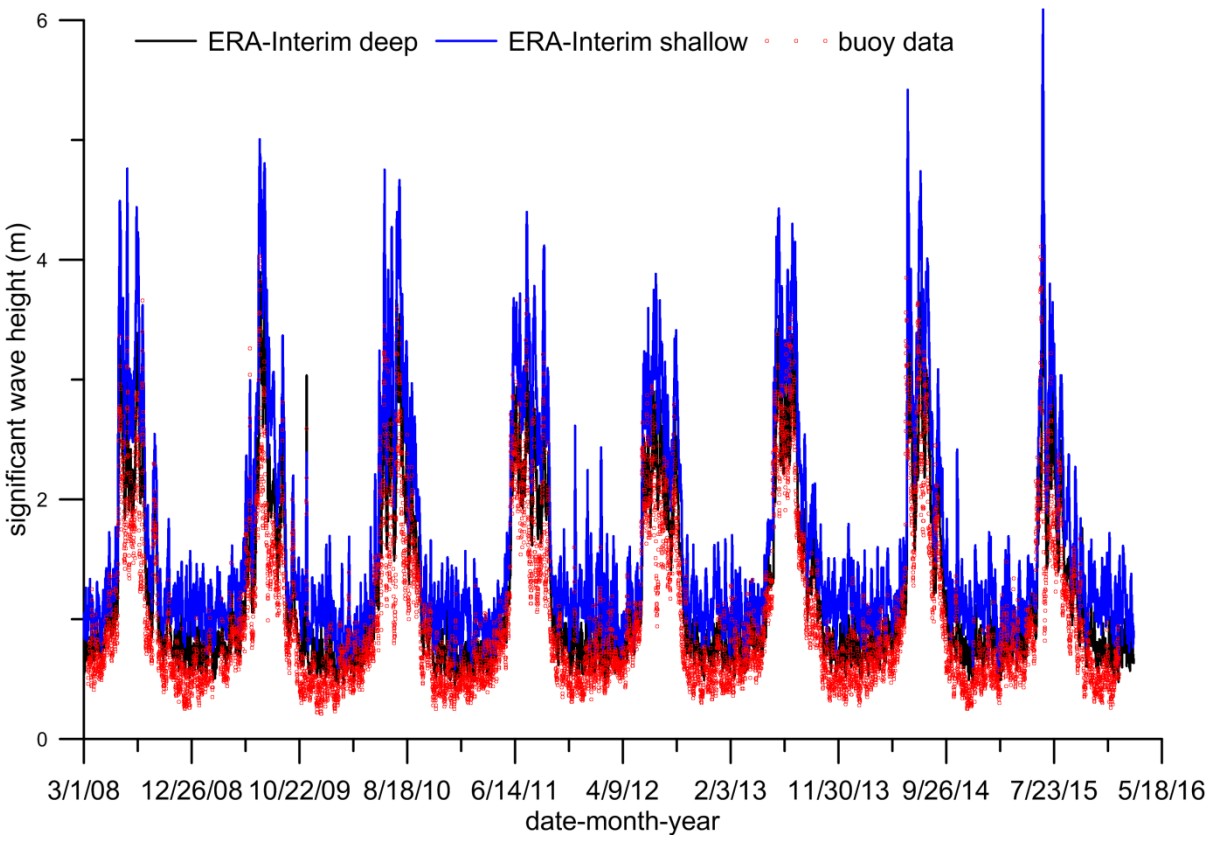

Figure 1: Time series plot of the significant wave height measured by buoy and from ERA-Interim reanalysis data at shallow and deep water

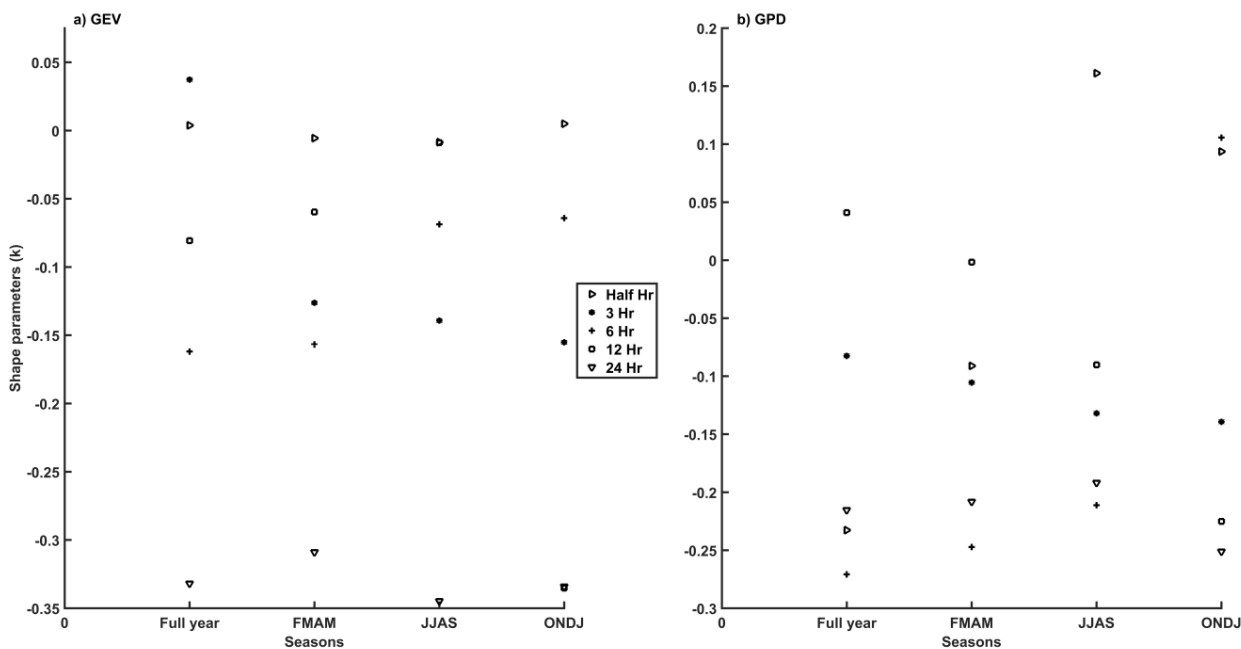

Figure 2: Estimated shape parameters for different seasonal data with different sampling intervals used in a) GEV and b) GPD model

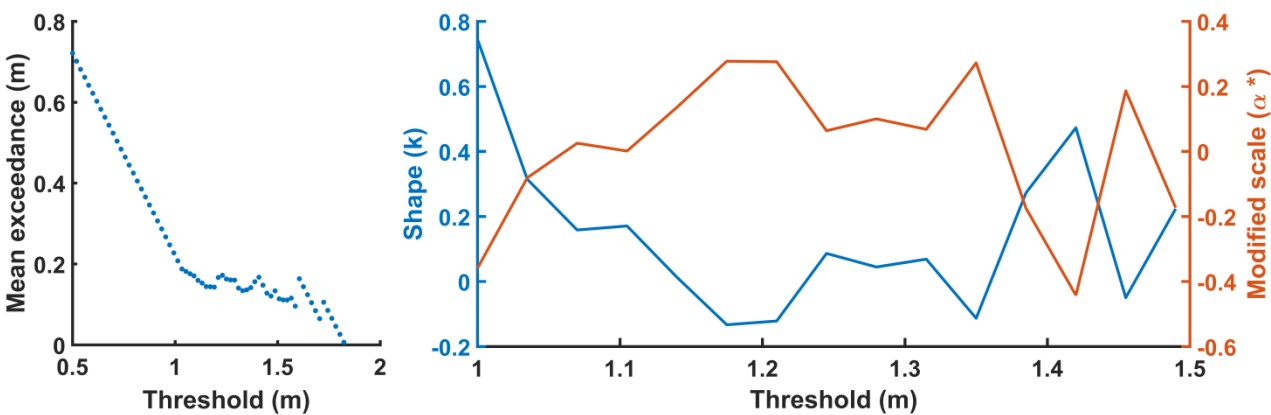

Figure 3: A Typical (a) SME and (b) PS plots used for selecting a range of thresholds required for POT analysis. In this particular case, a range of 1.19 m to 1.32 m was selected

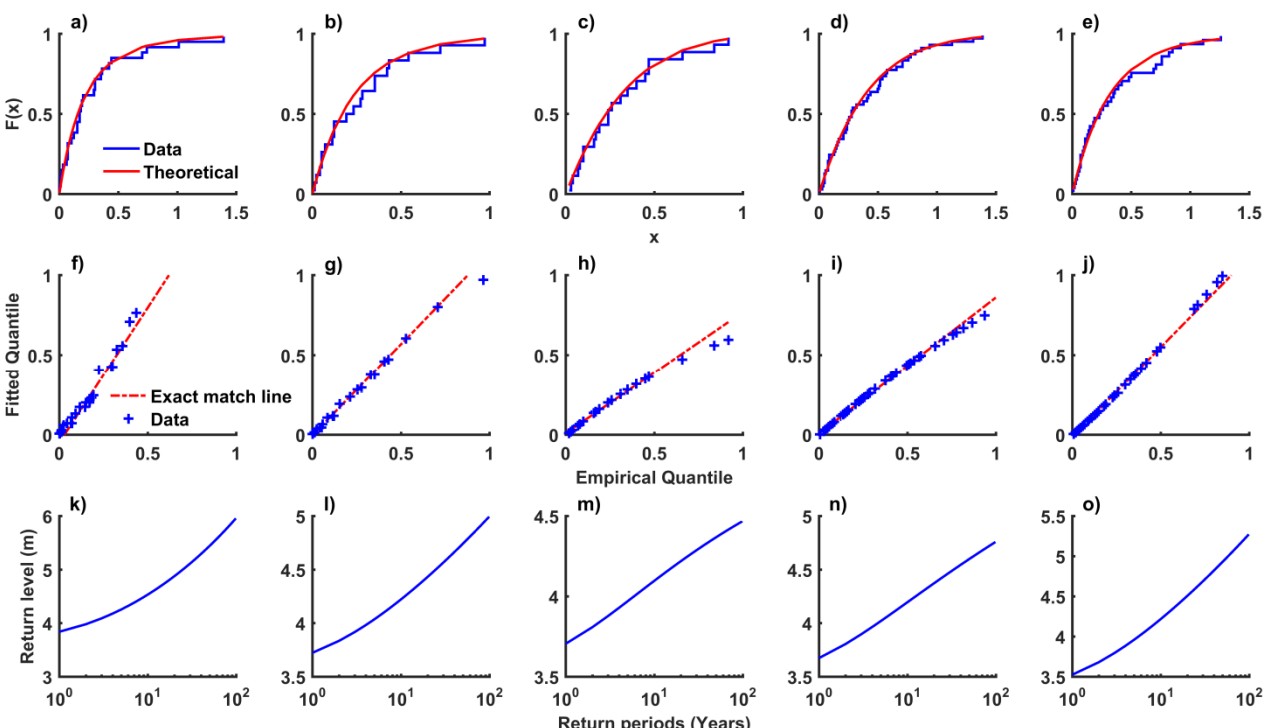

Figure 4: Figure corresponding to full year analysis. (a) to (e) is CDF plots for ½ hourly to 24 hourly data respectively, sub-figures, (f) to (g) are corresponding Q-Q plots and (k) to (o) are corresponding return levels estimated using GPD model

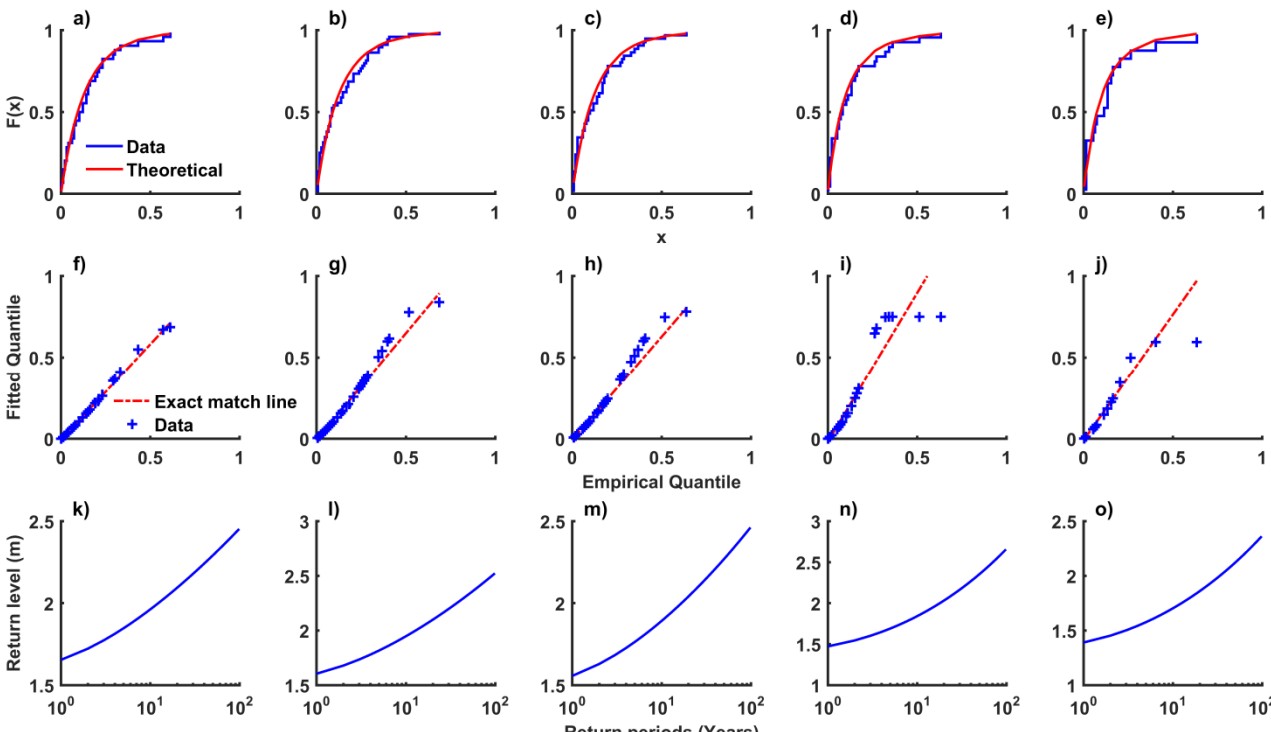

Figure 5: Same as in Figure 4 but corresponding to pre-monsoon season

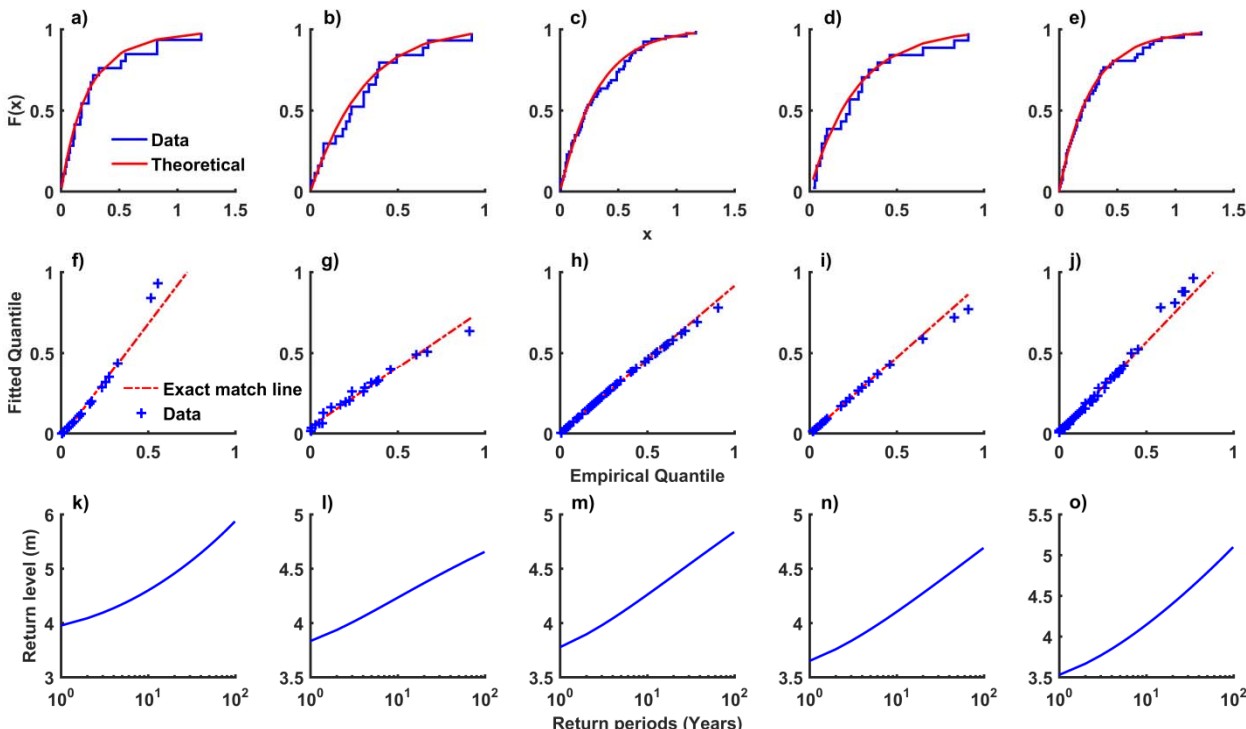

Figure 6: Same as in Figure 4 but corresponding to monsoon season

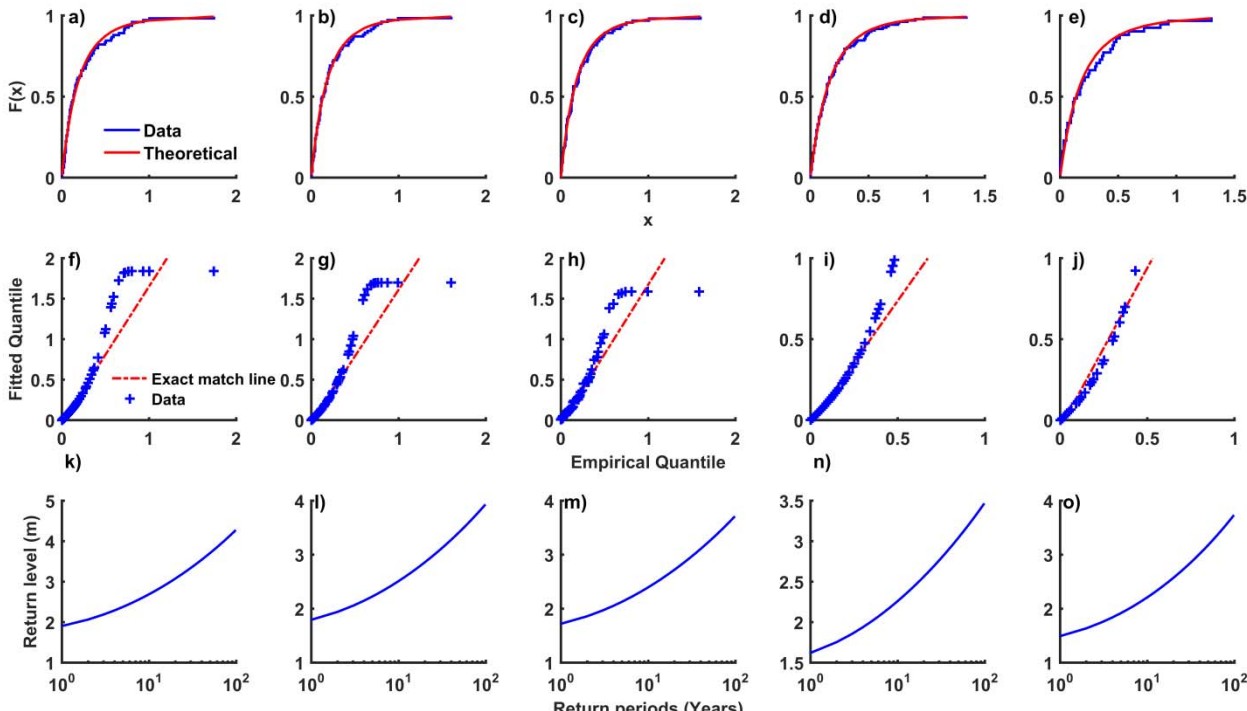

Figure 7: Same as in Figure 4 but corresponding to the post-monsoon season

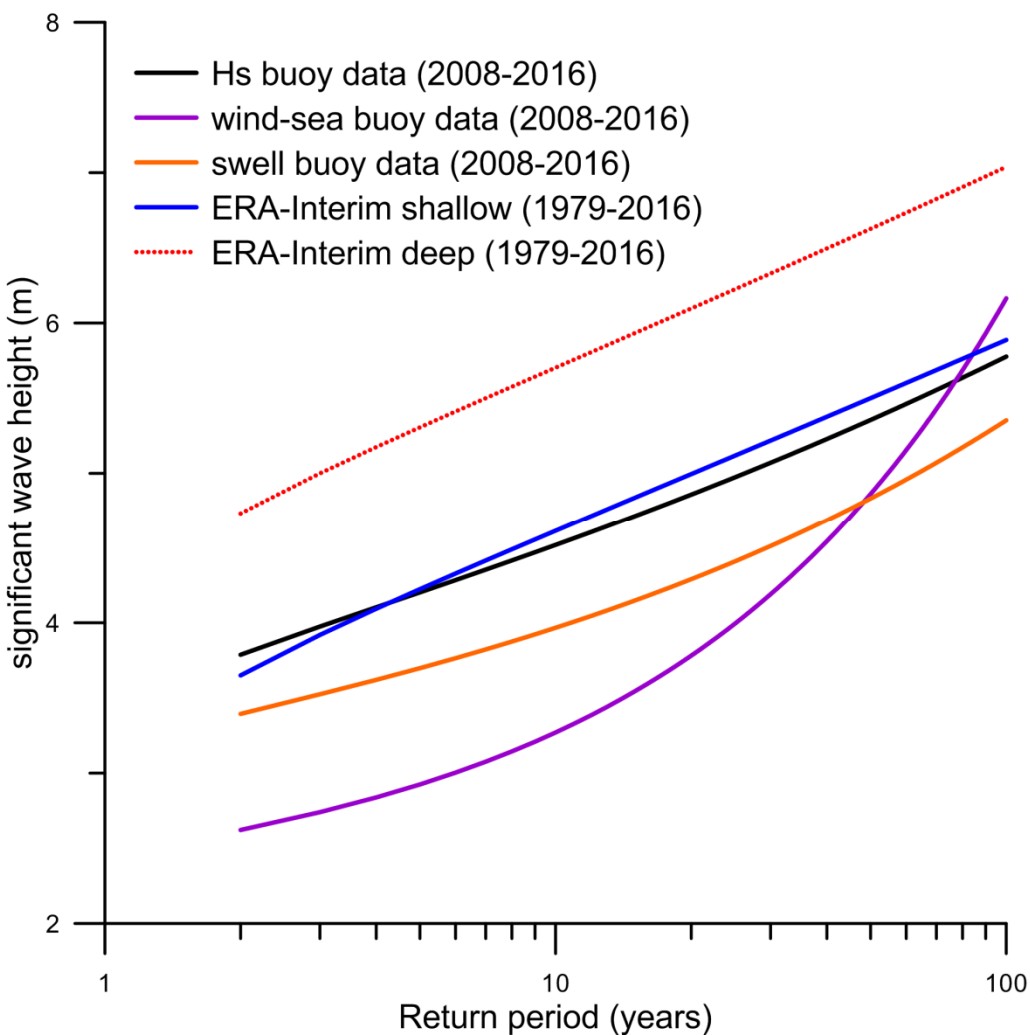

Figure 8: Return levels of significant wave heights for different return periods based on buoy data (2008-2016), ERA-Interim shallow and deep water (1979-2016) at 6-h interval by GEV model using annual maxima series

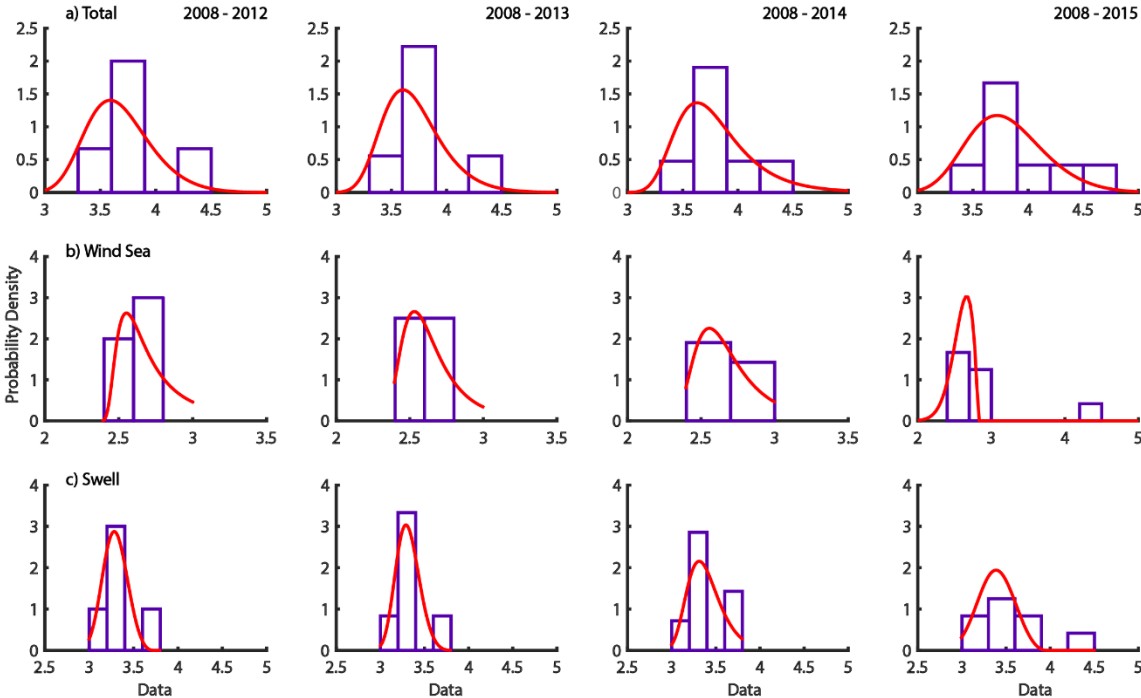

Figure 9: Density plots showing the probability for different wave height class. Total, wind-sea and swell Hs are presented in rows wise. Columns correspond to selected number of data points (5 to 8 years). The solid curve is the corresponding GPD fit

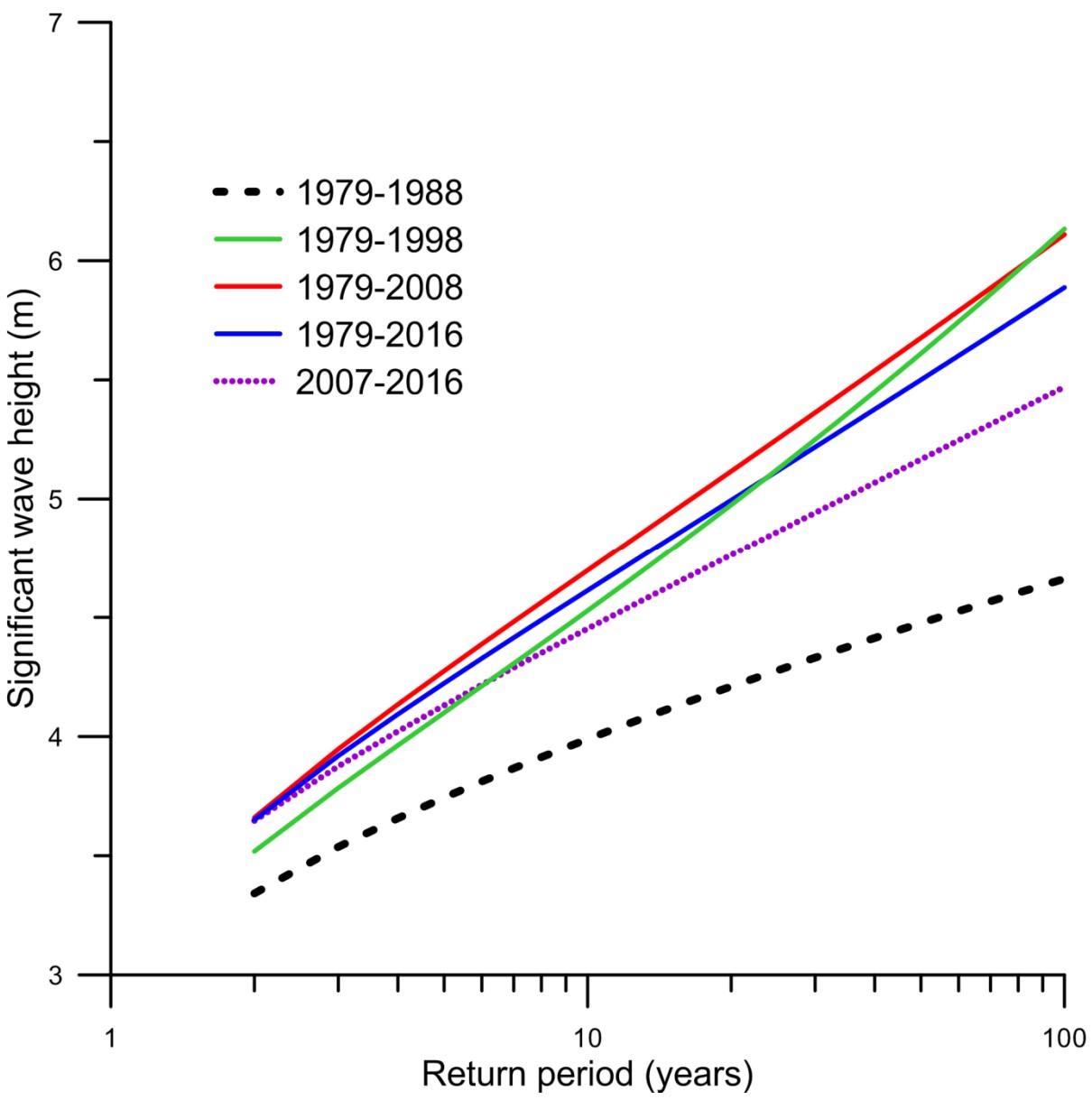

Figure 10. Return levels of significant wave heights for different return periods based on ERA-Interim shallow water data in different block years by GEV model using annual maxima series

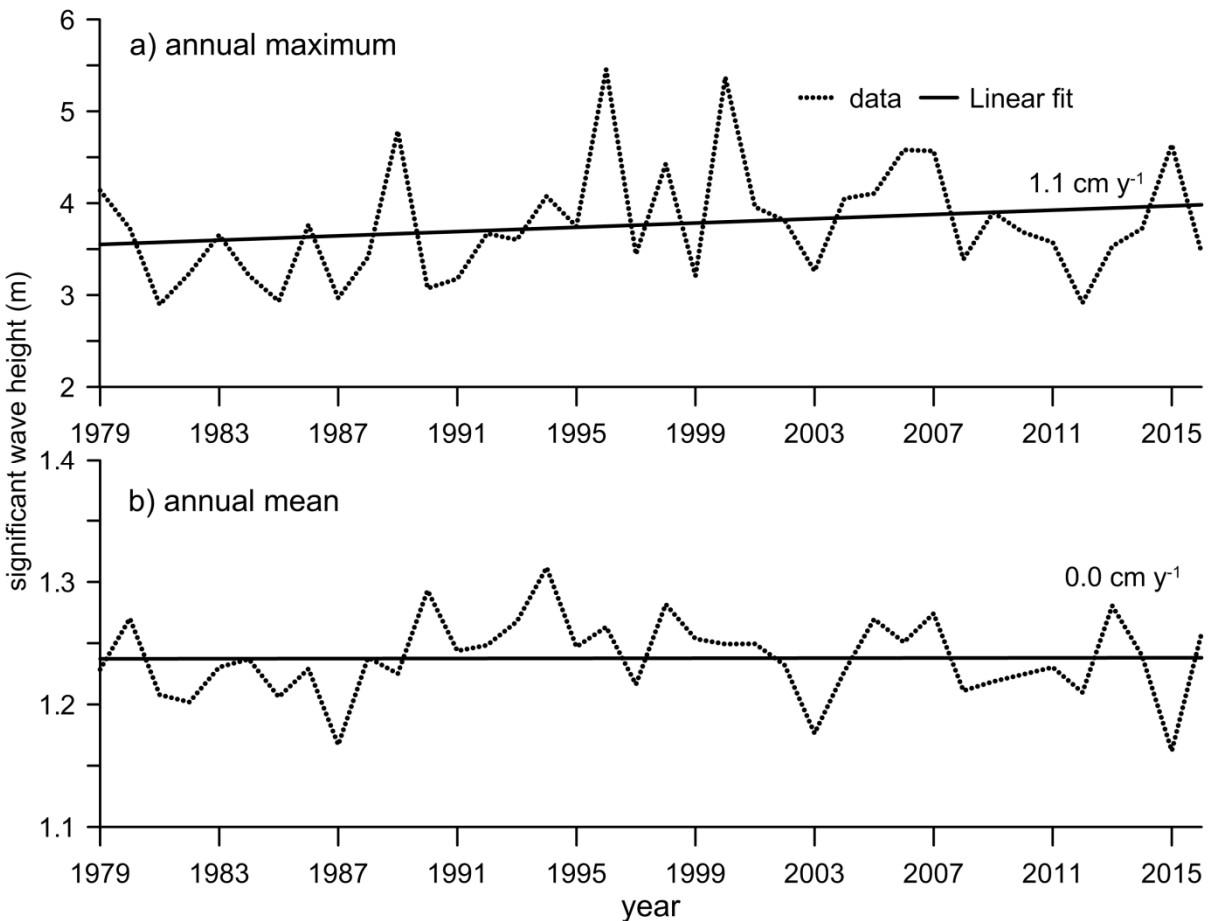

Figure 11. Variation of (a) annual maximum and (b) annual mean Hs at the shallow locations based on ERA-Interim data. The solid line indicates the trend in Hs during 1979 to 2016

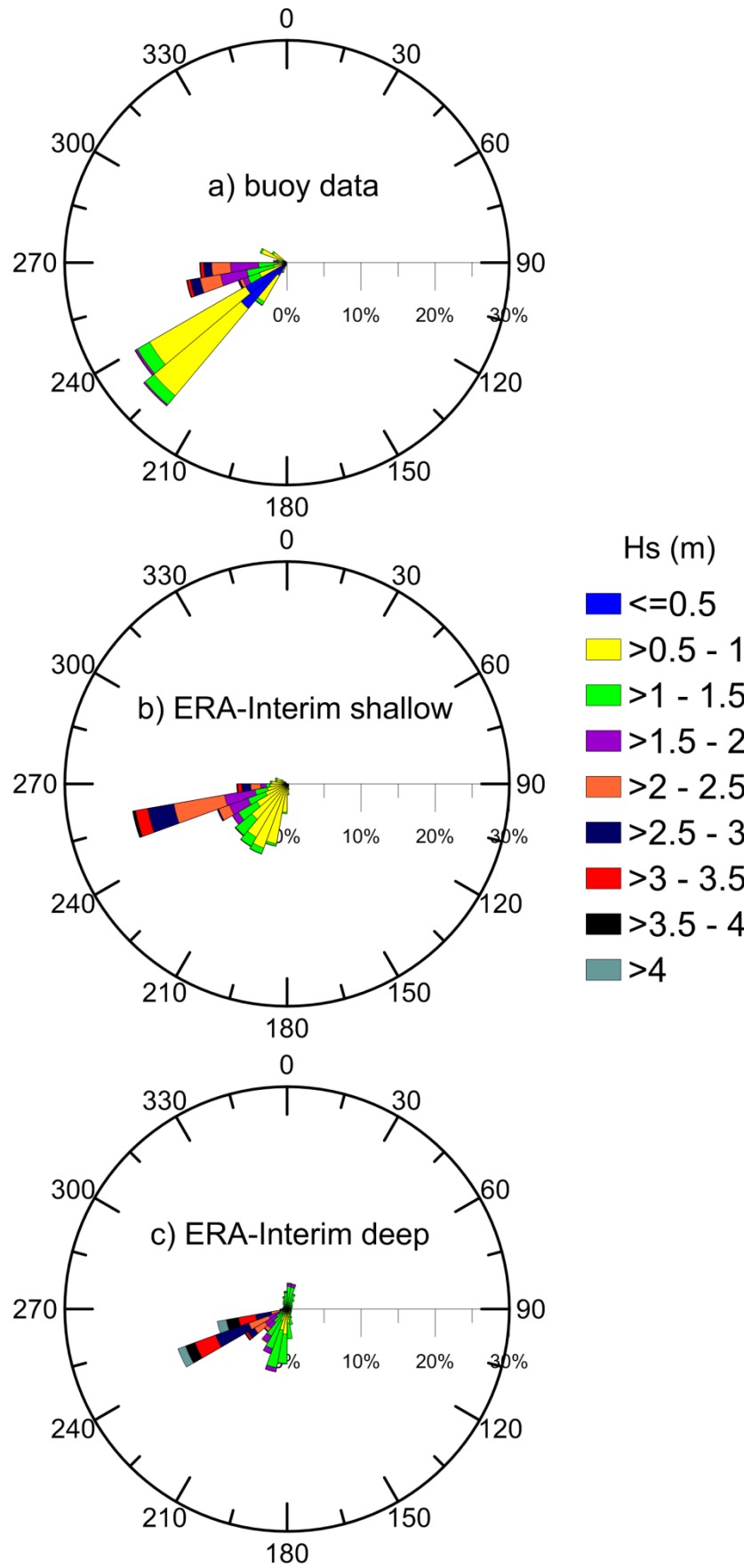

Figure 12: Wave rose plots during March 2008 to February 2016 based on the measured buoy data and the ERA-Interim reanalysis data at shallow and deep water locations

Table 1: The comparison of 50 and 100 year significant wave height return levels based on buoy, ERA-Interim shallow and ERA-Interim deep at 6-h interval along with data statistical parameters

| Distribution | Particulars | | Buoy | ERA-Interim shallow | ERA-Interim deep |
|---|---|---|---|---|---|
| | Location | | 74.391° E 14.304° N | 74.380° E 14.250° N | 69.250° E 14.250° N |
| GEV | Total number of data points Data period | | 11479 2008-2016 | 55520 1979-2016 | 55520 1979-2016 |
| | No. data points between different range | 1) >= 5m | 0 | 5 | 70 |
| | | 2) >= 4.5m & < 5m | 0 | 11 | 229 |
| | | 3) >= 3.5m & < 4.5m | 32 | 275 | 2846 |
| | | 4) >= 3m & < 3.5m | 137 | 1224 | 3263 |
| | DATA max (m) Second highest value (m) | | 4.11 4.03 | 5.45 5.37 | 7.13 6.09 |
| | DATA mean (m) | | 1.12 | 1.24 | 1.67 |
| | DATA std (m) | | 0.73 | 0.70 | 0.88 |
| | Return levels (m) | 50-y return period 100-y return period | 3.88 4.18 | 3.52 4.39 | 4.58 5.67 |
| GPD | No. data points between different range after decluster | 1) >= 5m | 0 | 2 | 15 |
| | | 2) >= 4.5m & < 5m | 0 | 4 | 42 |
| | | 3) >= 3.5m & < 4.5m | 9 | 46 | 253 |
| | | 4) >= 3m & < 3.5m | 23 | 133 | 235 |
| | Threshold (m) | | 3.19 | 3.50 | 4.41 |
| | Return levels (m) | 50-y return period 100-y return period | 4.36 4.46 | 5.55 6.18 | 6.69 7.28 |

Table 2: Different goodness of fittests used for selecting threshold values of POT analysis. H =0 indicates the test does not reject hypothesis at 5 % significance level (i.e., p-value > 0.05 or test statistics is less than critical value) and H=1 indicates hypothesis is rejected

| Seasons | Time Interval | $Hs_{max}$ (m) | Threshold (m) | KS test | | | | CM test | | | |
|---|---|---|---|---|---|---|---|---|---|---|---|
| | | | | p-value | Test statistics | Critical value | H | p-value | Test statistics | Critical value | H |
| Full Year | ½ h | 4.70 | 3.31 | 0.332 | 0.167 | 0.242 | 0 | 0.320 | 0.178 | 0.459 | 0 |
| | 3 h | 4.28 | 3.31 | 0.920 | 0.114 | 0.287 | 0 | 0.808 | 0.062 | 0.458 | 0 |
| | 6 h | 4.11 | 3.19 | 0.402 | 0.183 | 0.281 | 0 | 0.490 | 0.122 | 0.458 | 0 |
| | 12 h | 4.11 | 2.72 | 0.745 | 0.092 | 0.187 | 0 | 0.595 | 0.098 | 0.460 | 0 |
| | 24 h | 4.00 | 2.74 | 0.525 | 0.126 | 0.213 | 0 | 0.739 | 0.072 | 0.459 | 0 |
| FMAM | ½ h | 1.94 | 1.32 | 0.952 | 0.081 | 0.218 | 0 | 0.985 | 0.027 | 0.459 | 0 |
| | 3 h | 1.88 | 1.19 | 0.258 | 0.126 | 0.170 | 0 | 0.222 | 0.226 | 0.460 | 0 |
| | 6 h | 1.83 | 1.19 | 0.203 | 0.151 | 0.192 | 0 | 0.210 | 0.234 | 0.460 | 0 |
| | 12 h | 1.83 | 1.19 | 0.447 | 0.143 | 0.227 | 0 | 0.446 | 0.134 | 0.459 | 0 |
| | 24 h | 1.83 | 1.19 | 0.296 | 0.210 | 0.294 | 0 | 0.423 | 0.142 | 0.458 | 0 |
| JJAS | ½ h | 4.70 | 3.49 | 0.562 | 0.158 | 0.275 | 0 | 0.665 | 0.085 | 0.458 | 0 |
| | 3 h | 4.28 | 3.36 | 0.722 | 0.141 | 0.281 | 0 | 0.657 | 0.087 | 0.458 | 0 |
| | 6 h | 4.11 | 2.94 | 0.766 | 0.084 | 0.174 | 0 | 0.758 | 0.069 | 0.460 | 0 |
| | 12 h | 4.11 | 3.20 | 0.890 | 0.117 | 0.281 | 0 | 0.906 | 0.046 | 0.458 | 0 |
| | 24 h | 4.00 | 2.78 | 0.961 | 0.070 | 0.194 | 0 | 0.990 | 0.024 | 0.460 | 0 |
| ONDJ | ½ h | 2.81 | 1.06 | 0.131 | 0.123 | 0.144 | 0 | 0.193 | 0.247 | 0.460 | 0 |
| | 3 h | 2.61 | 1.00 | 0.247 | 0.106 | 0.142 | 0 | 0.307 | 0.183 | 0.460 | 0 |
| | 6 h | 2.59 | 0.98 | 0.488 | 0.092 | 0.151 | 0 | 0.451 | 0.133 | 0.460 | 0 |
| | 12 h | 2.18 | 0.84 | 0.197 | 0.102 | 0.129 | 0 | 0.350 | 0.166 | 0.461 | 0 |
| | 24 h | 2.18 | 0.87 | 0.195 | 0.155 | 0.196 | 0 | 0.207 | 0.237 | 0.460 | 0 |

Table 3: Table showing different parameters and corresponding RMSE of data and estimated CDF used during each data series analysis

| Seasons | Data | GPD | | | GEV | | | |
|---|---|---|---|---|---|---|---|---|
| | | k (m) | α (m) | RMSE (m) | k (m) | α (m) | β (m) | RMSE (m) |
| Full Year | ½ h | -0.233 | 0.208 | 0.222 | 0.004 | 0.416 | 2.462 | 0.023 |
| | 3 h | -0.091 | 0.232 | 0.028 | -0.006 | 0.381 | 2.439 | 0.008 |
| | 6 h | 0.161 | 0.346 | 0.110 | -0.008 | 0.418 | 2.223 | 0.004 |
| | 12 h | 0.094 | 0.420 | 0.102 | 0.005 | 0.416 | 2.206 | 0.020 |
| | 24 h | -0.082 | 0.314 | 0.071 | 0.037 | 0.458 | 2.015 | 0.060 |
| FMAM | ½ h | -0.105 | 0.130 | 0.037 | -0.126 | 0.115 | 1.134 | 0.090 |
| | 3 h | -0.132 | 0.125 | 0.078 | -0.139 | 0.104 | 1.143 | 0.098 |
| | 6 h | -0.139 | 0.123 | 0.077 | -0.155 | 0.099 | 1.147 | 0.100 |
| | 12 h | -0.271 | 0.095 | 0.167 | -0.162 | 0.108 | 0.998 | 0.125 |
| | 24 h | -0.247 | 0.099 | 0.082 | -0.157 | 0.114 | 0.872 | 0.142 |
| JJAS | ½ h | -0.184 | 0.216 | 0.124 | -0.069 | 0.298 | 2.782 | 0.088 |
| | 3 h | -0.046 | 0.280 | 0.068 | -0.069 | 0.274 | 2.786 | 0.074 |
| | 6 h | 0.041 | 0.328 | 0.051 | -0.081 | 0.288 | 2.583 | 0.118 |
| | 12 h | -0.002 | 0.265 | 0.042 | -0.060 | 0.281 | 2.598 | 0.065 |
| | 24 h | -0.090 | 0.267 | 0.083 | -0.009 | 0.312 | 2.423 | 0.007 |
| ONDJ | ½ h | -0.225 | 0.189 | 0.393 | -0.335 | 0.117 | 1.023 | 0.631 |
| | 3 h | -0.215 | 0.178 | 0.333 | -0.332 | 0.116 | 1.025 | 0.533 |
| | 6 h | -0.208 | 0.177 | 0.284 | -0.309 | 0.114 | 0.912 | 0.525 |
| | 12 h | -0.192 | 0.167 | 0.267 | -0.345 | 0.104 | 0.911 | 0.523 |
| | 24 h | -0.251 | 0.183 | 0.315 | -0.334 | 0.111 | 0.780 | 0.498 |

Table 4: Estimated return values corresponding to different seasons using total wave height (Hs) following GEV and GPD methods. Here GEV method follows initial distribution approach

| Seasons | DATA | GPD | | | GEV | | |
|---|---|---|---|---|---|---|---|
| | | 10 Years (m) | 50 Years (m) | 100 Years (m) | 10 Years (m) | 50 Years (m) | 100 Years (m) |
| Full Year | ½ h | 4.24 | 4.73 | 4.96 | 3.37 | 4.00 | 4.26 |
| | 3 h | 4.01 | 4.36 | 4.50 | 3.28 | 3.88 | 4.13 |
| | 6 h | 4.08 | 4.37 | 4.47 | 3.17 | 3.88 | 4.18 |
| | 12 h | 4.17 | 4.59 | 4.76 | 3.14 | 3.81 | 4.10 |
| | 24 h | 4.18 | 4.92 | 5.27 | 3.00 | 3.68 | 3.95 |
| FMAM | ½ h | 1.94 | 2.29 | 2.45 | 1.43 | 1.71 | 1.85 |
| | 3 h | 1.93 | 2.33 | 2.52 | 1.42 | 1.68 | 1.81 |
| | 6 h | 1.87 | 2.26 | 2.46 | 1.41 | 1.68 | 1.81 |
| | 12 h | 1.82 | 2.35 | 2.66 | 1.29 | 1.59 | 1.74 |
| | 24 h | 1.68 | 2.12 | 2.36 | 1.18 | 1.48 | 1.64 |
| JJAS | ½ h | 4.24 | 4.62 | 4.80 | 3.48 | 4.04 | 4.29 |
| | 3 h | 4.02 | 4.21 | 4.27 | 3.42 | 3.91 | 4.12 |
| | 6 h | 4.24 | 4.66 | 4.84 | 3.29 | 3.90 | 4.19 |
| | 12 h | 4.08 | 4.51 | 4.69 | 3.27 | 3.83 | 4.09 |
| | 24 h | 4.11 | 4.78 | 5.10 | 3.13 | 3.66 | 3.89 |
| ONDJ | ½ h | 2.64 | 3.69 | 4.28 | 1.41 | 1.96 | 2.30 |
| | 3 h | 2.46 | 3.41 | 3.93 | 1.41 | 1.95 | 2.28 |
| | 6 h | 2.35 | 3.23 | 3.71 | 1.28 | 1.77 | 2.07 |
| | 12 h | 2.22 | 3.03 | 3.47 | 1.27 | 1.77 | 2.09 |
| | 24 h | 2.16 | 3.16 | 3.74 | 1.15 | 1.67 | 2.00 |

Table 5: Return levels estimated by GEV model using total, wind-sea and swell data for different block maxima series

| DATA | Total Hs (m) | | | wind-sea Hs (m) | | | swell Hs (m) | | |
|---|---|---|---|---|---|---|---|---|---|
| | 10 years | 50 Years | 100 Years | 10 years | 50 Years | 100 Years | 10 years | 50 Years | 100 Years |
| Monthly Maxima | 3.21 | 5.28 | 6.02 | 2.45 | 3.43 | 3.88 | 2.92 | 4.77 | 5.72 |
| Two maximum values from each season | 3.66 | 5.58 | 6.56 | 2.68 | 3.78 | 4.29 | 3.31 | 5.07 | 5.95 |
| One maximum value from each season | 3.85 | 6.04 | 7.20 | 2.91 | 4.32 | 5.06 | 3.51 | 5.40 | 6.35 |
| Annual Maxima | 4.50 | 5.28 | 5.66 | 3.27 | 4.86 | 6.16 | 3.97 | 4.83 | 5.35 |

Table 6: Table showing the results of the case study. Standard deviation (STD) of each data series considered are provided, and percentage difference among the STD of each series with parent series (S0) are given in the brackets. Percentage difference in the corresponding return level estimation also shown in the brackets of respective return periods.

| Dataset | Series (Years) | Maximum observed (m) | Standard deviation (% difference) | Return levels | | |
|---|---|---|---|---|---|---|
| | | | | 10 Years (m) | 50 Years (m) | 100 Years (m) |
| Total | S1 (2008-2012) | 4.32 | 0.36 (21.8) | 4.24 (6.1) | 4.89 (8.6) | 5.20 (10.42) |
| | S2 (2008-2013) | 4.32 | 0.32 (32.7) | 4.17 (8.0) | 4.67 (13.1) | 4.90 (15.5) |
| | S3 (2008-2014) | 4.32 | 0.32 (34.5) | 4.23 (6.4) | 4.65 (13.5) | 4.83 (17.2) |
| | S0 | 4.70 | 0.45 | 4.50 | 5.28 | 5.66 |
| Wind-sea | S1 (2008-2012) | 2.80 | 0.13 (128.90) | 2.82 (14.81) | 2.88 (51.29) | 2.89 (72.30) |
| | S2 (2008-2013) | 2.80 | 0.14 (125.06) | 2.81 (15.00) | 2.95 (48.96) | 3.00 (69.16) |
| | S3 (2008-2014) | 2.89 | 0.16 (114.08) | 2.89 (12.35) | 3.05 (45.80) | 3.11 (66.00) |
| | S0 | 4.29 | 0.60 | 3.27 | 4.86 | 6.16 |
| Swell | S1 (2008-2012) | 3.47 | 0.23 (48.17) | 3.65 (8.23) | 4.16 (14.93) | 4.45 (18.36) |
| | S2 (2008-2013) | 3.47 | 0.20 (58.53) | 3.62 (9.18) | 4.01 (18.53) | 4.22 (23.56) |
| | S3 (2008-2014) | 3.47 | 0.22 (50.80) | 3.71 (6.62) | 4.05 (17.53) | 4.21 (23.97) |
| | S0 | 4.28 | 0.37 | 3.97 | 4.83 | 5.35 |

Table 7: The percentage of time the waves in shallow, intermediate and deep water regime in different years along with the mean wave period and mean peak wave period

| Year | Mean wave period (s) | Criteria based on ratio of water depth and wave length corresponding to mean wave period | | | Mean peak wave period (s) | Criteria based on ratio of water depth and wave length corresponding to peak wave period | | |
|---|---|---|---|---|---|---|---|---|
| | | Shallow water | Intermediate water | Deep water | | Shallow water | Intermediate water | Deep water |
| 2008-2009 | 5.5 | 0 | 98.7 | 1.3 | 12.1 | 1.0 | 98.9 | 0.1 |
| 2009-2010 | 5.6 | 0 | 98.3 | 1.6 | 12.0 | 0.5 | 99.3 | 0.2 |
| 2010-2011 | 5.4 | 0 | 97.5 | 2.5 | 11.7 | 0.6 | 99.2 | 0.2 |
| 2011-2012 | 5.7 | 0 | 99.5 | 0.5 | 11.9 | 0.9 | 98.5 | 0.6 |
| 2012-2013 | 5.5 | 0 | 99.4 | 0.6 | 12.0 | 0.3 | 99.6 | 0.1 |
| 2013-2014 | 5.0 | 0 | 95.0 | 5.0 | 11.8 | 1.4 | 97.8 | 0.8 |
| 2014-2015 | 5.7 | 0 | 98.7 | 1.3 | 12.6 | 1.8 | 98.1 | 0.1 |
| 2015-2016 | 5.5 | 0 | 98.0 | 2.0 | 12.3 | 0.8 | 99.0 | 0.2 |