# Peer review of "Variations in return value estimate of ocean surface waves - a study based on measured buoy data and ERA-Interim reanalysis data"

_Natural Hazards and Earth System Sciences, 2017_

## Referee Comment (RC1) · Anonymous Referee #1 · 17 Jul 2017

Review of the manuscript nhess – 2017-165 "Variations in return value estimate of ocean surface waves a study based on measured buoy data" written by T. Muhammed Naseef and V. Sanil Kumar. My recommendation is that the paper is accepted but with major changes. In my view this paper is very interesting and the topic is of special interest for the scientific and engineering communities. However, there are a few outstanding points to mention. In the first instance, with only 8 years of wave data, the uncertainty is too great to estimate extreme values for a 100 year return period. Did you perform any sensitivity analysis? Waves were measured in a water depth of 9 m. In principle the data collected is strongly influenced by the wave period. I am almost sure that in the case of very severe storms there is a high probability that some waves

arrive very steep or broken. Did you take into account these processes for the characterization of extreme values? For the most extreme conditions, in addition to Hs, it is also very important to consider the wave period and the duration of the storm. It is not explained in the paper how the Sea and Swell were separated from the original records. Very useful work has been carried out recently (see references) which does not seem to have been consulted. I strongly suggest the author check the state of the art. Why was the criteria used to separate the seasons that of months? It might be more reliable to use the atmospheric pressure values, as the Monson season is not always the same. It is the first time that I have read in a paper (Acknowledgements) that a contract is xxxx. My major concerns with the paper are: the length of the wave record, the fact that the analysis does not consider the physical processes involved, nor the effect of the wave period, direction of incidence of the waves and duration of the storms.

References Solari, S., & Losada, M. A. (2011). Non‐stationary wave height climate modeling and simulation. Journal of Geophysical Research: Oceans, 116(C9). Solari, S., & Losada, M. Á. (2012). Unified distribution models for met-ocean variables: Application to series of significant wave height. Coastal Engineering, 68, 67-77. Solari, S., Egüen, M., Polo, M. J., & Losada, M. A. (2017). Peaks Over Threshold (POT): A methodology for automatic threshold estimation using goodness of fit p‐value. Water Resources Research, 53(4), 2833-2849. Solari, S., & Losada, M. A. (2015). Statistical Methods for Risk Assessment of Harbor and Coastal Structures. In Design of Coastal Structures and Sea Defenses (pp. 215-272). Folgueras, P., Solari, S., Mier-Torrecilla, M., Doblaré, M., & Losada, M. Á. (2016). The extended Davenport peak factor as an extreme-value estimation method for linear combinations of correlated non-Gaussian random variables. Journal of Wind Engineering and Industrial Aerodynamics, 157, 125-139.
* * *

---

## Referee Comment (RC2) · Anonymous Referee #2 · 17 Jul 2017

This paper provides an extreme value analysis of the significant wave height (Hs) in a buoy of India spanning for 8 years (2008-2016). It includes a detail comparison on the estimation of return levels using different extreme distributions (GEV vs GPD) and maxima selection (block maxima and peak over threshold), a set of statistical test to analyze threshold selection and model fit, and a sensitivity analysis considering different spanning periods. The analysis is performed to the total Hs, Wind Sea and swell, and, in the temporary scale they consider the annual and seasonal wave climate. The paper carries out a detailed but conventional analysis of extreme wave climate comparing different methods and tools. The main drawback is that the analysis is performed to an 8-year record of data, which is a very short period of time to analyze extreme behavior.

[Figure]

I miss a discussion about that in the manuscript and comments/references/analysis of interannual and decadal variability which can influence wave climate in such short periods of time. I find very cheeky and uncertain to obtain 100-year return levels extrapolating from an 8-year record. Besides nothing is told about the influence of water depth (9 m) that can also condition the extreme wave climate. For me it is not clear if the aim of the paper is to show a statistical analysis with a set of test, tools and comparisons or if it tries to provide useful information and knowledge about the extreme wave climate in India. In the first case, although a complete EVA is posed, I do not consider it in the cutting edge. In the second case (which I consider the really interesting thing for the NHESS topics), I consider the use of an 8-year record buoy not appropriate for providing conclusions about extreme wave climate. I encourage the authors to carry out a similar analysis over a longer record buoy, satellite or reanalysis data. Extra explanations about how Wind Sea and swell are considered, the influence of water depth, influence of wave direction in wind sea and swell, or interannual variability in the area should be considered. I recommend that the manuscript only be considered for publication after major revision.
* * *

---

## Author Comment (AC1) · 24 Aug 2017

Thanks for the suggestions. We have now revised the manuscript considering all the suggestions. Since the measured buoy data is for a period of 8 years only, the Hs data from the ERA-Interim (Dee et al. 2011), the global atmospheric reanalysis product of the European Centre for Medium Range Weather Forecast (ECMWF) for 38 years (from 1979 to 2016) is now used to evaluate the wave height with different return period in the shallow (water depth ~20m) and the deep water. The shallow region is close to the buoy location and the deep water location is at a water depth of ~4000 m (Table 1). ERA-Interim used in the study has a spatial resolution is 0.125 X 0.125 deg and a

temporal resolution of 6-h.

Dee, D. P., Uppala, S. M., Simmons, A. J., Berrisford, P., Poli, P., Kobayashi, S., Andrae, U., Balmaseda, M. A., Balsamo, G., Bauer, P., Bechtold, P., Beljaars, A. C. M., van de Berg, L., Bidlot, J., Bormann, N., Delsol, C., Dragani, R., Fuentes, M., Geer, A. J., Haimberger, L., Healy, S. B., Hersbach, H., Hólm, E. V., Isaksen, L., Kållberg, P., Köhler, M., Matricardi, M., McNally, A. P., Monge-Sanz, B. M., Morcrette, J.-J., Park, B.-K., Peubey, C., de Rosnay, P., Tavolato, C., Thépaut, J.-N., and Vitart, F.: The ERA-Interim reanalysis: Configuration and performance of the data assimilation system, Q. J. Roy. Meteor. Soc., 137, 553–597, 2011.

A section on "influence of length of wave data on estimated significant wave height return value" is now added. Since the data is collected at 9-m water depth, most of the time it is in intermediate wave regime. The data on mean wave period and the peak wave period are now added. A table is now added to show the percentage of time the waves are in intermediate, shallow and deep water regime based on relative depth (ratio of the water depth and the wave length) and the results are discussed. The wave breaking aspects are now covered. The methodology on wind-sea and swell separation are also added. The waves in the west coast of India are strongly season depended. Hence, we have considered seasons based on months. For the study location, the storm induced wave heights are less than the monsoon induced waves. June first week is the onset of Indian summer monsoon and the maximum Hs in the study area is due to monsoon influence and in all years it is during June to September.

The recent works as suggested are now cited.

Now we have added a wave rose plot showing the wave direction of measured data and the ERA-Interim data and discussed in the paper.

Other minor correction related to contribution number is corrected.

[Figure]

2017-165, 2017.

[Figure]

Figure 8: Return levels of significant wave heights for different return periods based on buoy data (2008-2016), ERA-Interim shallow and deep water (1979-2016) at 6-h interval by GEV model using annual maxima series

26

**Fig. 1.** Return levels of significant wave heights for different return periods based on buoy data (2008-2016), ERA-Interim shallow and deep water (1979-2016) at 6-h interval by GEV model using annual maxima s

---

## Author Comment (AC2) · 24 Aug 2017

Thanks for the suggestions. We have now carried out a major revision of the manuscript considering all the suggestions.

Since the measured buoy data is for a period of 8 years only, the Hs data from the ERA-Interim (Dee et al. 2011), the global atmospheric reanalysis product of the European Centre for Medium Range Weather Forecast (ECMWF) for 38 years (from 1979 to 2016) is used to evaluate the wave height with 100 and 50-year return period in the shallow (water depth ∼20m) and the deep water. The shallow region is close to the buoy location and the deep water location is at a water depth of ∼4000 m (Table

1). ERA-Interim used in the study has a spatial resolution is 0.125 X 0.125ïĆř and a temporal resolution of 6 h.

Now we have added a Figure showing the inter-annual variations in the annual mean and annual maximum Hs based on the ERA-Interim data covering 38 years and discussed the inter-annual variations (see Figure attached).

Influence of water depth is studied based on the relative depth (ration of water depth to wave length).

Extra explanations about how Wind Sea and swell are considered, the influence of water depth, influence of wave direction and the inter-annual variability in the area are now included. A figure showing the inter-annual variability is also added.
* * *
Figure 11. Variation of (a) annual maximum and (b) annual mean Hs at the shallow locations
based on ERA-Interim data. The solid line indicates the trend in Hs during 1979 to 2016

**Fig. 1.** Variation of (a) annual maximum and (b) annual mean Hs at the shallow locations based
on ERA-Interim data. The solid line indicates the trend in Hs during 1979 to 2016

---

## Author Response (AR1)

**Response to Anonymous Referee #1**

Review of the manuscript nhess – 2017-165 "Variations in return value estimate of ocean surface waves a study based on measured buoy data" written by T. Muhammed Naseef and V. Sanil Kumar. My recommendation is that the paper is accepted but with major changes. In my view this paper is very interesting and the topic is of special interest for the scientific and engineering communities. However, there are a few out standing points to mention. In the first instance, with only 8 years of wave data, the uncertainty is too great to estimate extreme values for a 100 year return period. Did you perform any sensitivity analysis? Waves were measured in a water depth of 9 m. In principle the data collected is strongly influenced by the wave period. I am almost sure that in the case of very severe storms there is a high probability that some waves arrive very steep or broken. Did you take into account these processes for the characterization of extreme values? For the most extreme conditions, in addition to Hs, it is also very important to consider the wave period and the duration of the storm. It is not explained in the paper how the Sea and Swell were separated from the original records. Very useful work has been carried out recently (see references) which does not seem to have been consulted. I strongly suggest the author check the state of the art. Why was the criteria used to separate the seasons that of months? It might be more reliable to use the atmospheric pressure values, as the Monson season is not always the same. It is the first time that I have read in a paper (Acknowledgements) that a contract is xxxx. My major concerns with the paper are: the length of the wave record, the fact that the analysis does not consider the physical processes involved, nor the effect of the wave period, direction of incidence of the waves and duration of the storms.

References Solari, S., & Losada, M. A. (2011). Non˘A˘ R˘ stationary wave height climate modeling and simulation. Journal of Geophysical Research: Oceans, 116(C9). Solari, S., & Losada, M. Á. (2012). Unified distribution models for met-ocean variables: Ap- plication to series of significant wave height. Coastal Engineering, 68, 67-77. Solari, S., Egüen, M., Polo, M. J., & Losada, M. A. (2017). Peaks Over Threshold (POT): A methodology for automatic threshold estimation using goodness of fit p˘A˘ R˘ value. Water Resources Research, 53(4), 2833-2849. Solari, S., & Losada, M. A. (2015). Statistical Methods for Risk Assessment of Harbor and Coastal Structures. In Design of Coastal Structures and Sea Defenses (pp. 215-272). Folgueras, P., Solari, S., Mier- Torrecilla, M., Doblaré, M., & Losada, M. Á. (2016). The extended Davenport peak factor as an extreme-value estimation method for linear combinations of correlated non-Gaussian random variables. Journal of Wind Engineering and Industrial Aerody- namics, 157, 125-139

**Reply:**

Thanks for the suggestions. We have now revised the manuscript considering all the suggestions and the corrected manuscript in track-changes is attached. Since the measured buoy data is for a period of 8 years only, the Hs data from the ERA-Interim (Dee et al. 2011), the global atmospheric reanalysis product of the European Centre for Medium Range Weather Forecast (ECMWF) for 38 years (from 1979 to 2016) is now used to evaluate the wave height with different return period in the shallow (water depth ~20m) and the deep water. The shallow region is close to the buoy location and the deep water location is at a water depth of ~4000 m (Table 1). ERA-Interim used in the study has a spatial resolution is 0.125 X 0.125° and a temporal resolution of 6 h.

Dee, D. P., Uppala, S. M., Simmons, A. J., Berrisford, P., Poli, P., Kobayashi, S., Andrae, U., Balmaseda, M. A., Balsamo, G., Bauer, P., Bechtold, P., Beljaars, A. C. M., van de Berg, L.,

Bidlot, J., Bormann, N., Delsol, C., Dragani, R., Fuentes, M., Geer, A. J., Haimberger, L., Healy, S. B., Hersbach, H., Hólm, E. V., Isaksen, L., Kållberg, P., Köhler, M., Matricardi, M., McNally, A. P., Monge-Sanz, B. M., Morcrette, J.-J., Park, B.-K., Peubey, C., de Rosnay, P., Tavolato, C., Thépaut, J.-N., and Vitart, F.: The ERA-Interim reanalysis: Configuration and performance of the data assimilation system, Q. J. Roy. Meteor. Soc., 137, 553–597, 2011.

A section on "influence of length of wave data on estimated significant wave height return value" is now added (section 3.3). Since the data is collected at 9-m water depth, most of the time it is in intermediate wave regime. The data on mean wave period and the peak wave period are now added. A table is now added to show the percentage of time the waves are in intermediate, shallow and deep water regime based on relative depth (ratio of the water depth and the wave length) and the results are discussed. The wave breaking aspects are now covered.

The methodology on wind-sea and swell separation is also added (section 2.1).

The waves in the west coast of India are strongly season depended. Hence, we have considered seasons based on months. For the study location, the storm induced wave heights during the non-monsoon period are less than the monsoon induced waves. June first week is the onset of Indian summer monsoon and the maximum Hs in the study area is due to monsoon influence and in all years it is during June to September. These are now added in the manuscript under section 3.3.

The recent works as suggested are now cited.

Now we have added a wave rose plot showing the wave direction of measured data and the ERA-Interim data and discussed in the paper (Figure 12).

Other minor correction related to contribution number is corrected.

**Response to Anonymous Referee #2**

This paper provides an extreme value analysis of the significant wave height (Hs) in a buoy of India spanning for 8 years (2008-2016). It includes a detail comparison on the estimation of return levels using different extreme distributions (GEV vs GPD) and maxima selection (block maxima and peak over threshold), a set of statistical test to an- alyze threshold selection and model fit, and a sensitivity analysis considering different spanning periods. The analysis is performed to the total Hs, Wind Sea and swell, and, in the temporary scale they consider the annual and seasonal wave climate. The pa- per carries out a detailed but conventional analysis of extreme wave climate comparing different methods and tools. The main drawback is that the analysis is performed to an 8-year record of data, which is a very short period of time to analyze extreme behavior.

I miss a discussion about that in the manuscript and comments/references/analysis of interannual and decadal variability which can influence wave climate in such short periods of time. I find very cheeky and uncertain to obtain 100-year return levels extrapolating from an 8-year record. Besides nothing is told about the influence of water depth (9 m) that can also condition the extreme wave climate. For me it is not clear if the aim of the paper is to show a statistical analysis with a set of test, tools and comparisons or if it tries to provide useful information and knowledge about the extreme wave climate in India. In the first case,

although a complete EVA is posed, I do not consider it in the cutting edge. In the second case (which I consider the really interesting thing for the NHESS topics), I consider the use of an 8-year record buoy not appropriate for providing conclusions about extreme wave climate. I encourage the authors to carry out a similar analysis over a longer record buoy, satellite or reanalysis data. Extra explanations about how Wind Sea and swell are considered, the influence of water depth, influence of wave direction in wind sea and swell, or interannual variability in the area should be considered.

I recommend that the manuscript only be considered for publication after major revision

**Reply:**

Thanks for the suggestions. We have now carried out a major revision of the manuscript considering all the suggestions and the corrected manuscript in track-changes is attached.

Since the measured buoy data is for a period of 8 years only, the Hs data from the ERA-Interim (Dee et al. 2011), the global atmospheric reanalysis product of the European Centre for Medium Range Weather Forecast (ECMWF) for 38 years (from 1979 to 2016) is used to evaluate the wave height with 100 and 50-year return period in the shallow (water depth ~20m) and the deep water. The shallow region is close to the buoy location and the deep water location is at a water depth of ~4000 m (Table 1). ERA-Interim used in the study has a spatial resolution is 0.125 X 0.125° and a temporal resolution of 6 h.

Now we have added a Figure showing the inter-annual variations in the annual mean and annual maximum Hs based on the ERA-Interim data covering 38 years (Figure 11) and discussed the inter-annual variations.

Influence of water depth is studied based on the relative depth (ratio of water depth to wave length) in section 3.4.

Extra explanations about how Wind Sea and swell are considered, the influence of water depth, influence of wave direction and the inter-annual variability in the area are now included. A figure showing the inter-annual variability is also added (Figure 11). We have discussed the influence of storm on the 100-year Hs value (section 3.3).

[revised manuscript text omitted]